

# Exploring Limiting Factors of Wear in Pitch Bearings of Wind Turbines with Real Scale Tests

Karsten Behnke and Florian Schleich

Fraunhofer IWES, Am Schleusengraben 22, Hamburg, 21029, Germany

*Correspondence to*: Karsten Behnke (Karsten.Behnke@iwes.fraunhofer.de)

**Abstract.** Oscillating movements under load can cause wear in rolling bearings. Blade bearings of wind turbines are subject to both. To know how to avoid wear in these bearings is important since they ensure the operational safety of the turbine. Oscillations of blade bearings vary in load, speed, and amplitude. The objective of this work is to find limits of these operating

parameters with regards to wear occurrence. To this end several tests with real size bearings were carried out. The test parameters are based on typical operating conditions of a reference turbine. The size of the bearings and the test parameters differ from other published test results for oscillating bearings.

The test results show that wear occurs for every tested combination of load, speed, and amplitude of a steady oscillating movement. Even if the wear characteristics differ between tests, each of them resulted in wear. Hence, no wear limits can be

defined within typical operating conditions of a wind turbine below which wear does not occur.

Tests with a discontinuity in the steady oscillation movement, however, did not result in wear. Such discontinuities can be longer movements embedded in steady oscillations. They are characteristic to wind turbine operation, where longer movements are a reaction to wind gusts.

## 1 Introduction

Blade bearings allow the rotor blades of a wind turbine to rotate about their longitudinal axes, which is also called pitching. Thus, power and load control of the turbine as well as emergency stops are possible (Hau, 2017). The blade bearing reliability is important for the function and operational safety of a wind turbine. Loads and the pitch movements form the key operating conditions which influence the bearing reliability. Typical pitch movements are oscillations with a small amplitude. A continuous pitch control (CPC) turns all blades simultaneously by the same angle to adapt the power output (Burton et al.,

2021). An individual pitch control (IPC) turns each blade individually to reduce turbine fatigue loads (Bossanyi, 2003). Both controller types lead to different number of oscillations with different amplitudes and velocities (Stammler et al., 2018a). The acting forces from the wind and the blade design are the main drivers for the loads on a blade bearing (Harris et al., 2009). The bearing design, like the rolling element type or the number of rolling elements as well as the raceway geometry determine the loads in the rolling contact.



Stammler et al. considered the influence of an IPC controller on blade bearings (Stammler et al., 2019). They showed that the IPC controller leads to a higher pitch activity, which effects the damage mechanisms. One failure mode for a bearing is wear on the raceways. In contrast to raceway fatigue, which is inevitable after sufficient load cycles, the development of wear depends on the operating conditions like amplitude, cycles, speed, and load. Hence, Stammler et al. could not draw a conclusion with respect to the influence of the influence of an IPC on wear (Stammler et al., 2019).

Wear can lead to a significantly increased friction torque, and it can cause a bearing failure or trigger other damage modes like rolling contact fatigue. But wear is a damage that can be avoided or kept in a harmless state (Harris and Kotzalas, 2006), if critical conditions are avoided. There are many publications regarding wear in rolling bearings. To the knowledge of the authors, only a few of them refer explicitly to blade bearing application and include tests with bearings to increase the knowledge about the relation between operation parameters of a wind turbine and wear (Becker, 2012; Schwack et al., 2020;

Wandel S., and Bartschat B., 2021; Schwack et al., 2021; Stammler, 2020). The authors use different bearing sizes and types according to the available test infrastructure. Furthermore, they use different greases and different test parameters. All tests were done with static axial loads. A way to compare wear tests is the dimensionless x/2b ratio. It relates the travel of the rolling element during a half cycle (x) to the width of the Hertzian contact (2b). It is independent of bearing sizes.

Becker developed a test method to test greases under oscillating conditions. He used four-point bearings with an inner diameter

of 60 mm and a contact pressure of 3 GPa. During the test he varied the temperature and added salt-water to force corrosion, he tested six different greases (Becker, 2012). Neither the oscillation amplitude nor the x/2b ratio is published. Schwack et al. also tested six different greases. They considered three different x/2b ratios (0.9, 13.3, and 29.1) with constant contact pressure of 1.9 GPa for tests with angular contact ball bearings. The bearings have an inner diameter of 40 mm. They stated that the greases must fit to the application and that one grease could prevent wear for some conditions but fail for others. Furthermore,

they were not able to produce wear for the highest x/2b ratio (Schwack et al., 2020). Schwack et al. also tested two bearing types with an outer diameter of 750 mm and a contact pressure of 2.0 GPa. They differed between two x/2b ratios (2.67 and 11.44) and were able to create wear for both bearings and for both x/2b values (Schwack et al., 2021). So far, all test bearings have a diameter lower than one meter. Stammler was the first who published a wear test with a real scale blade bearing. He compared tests of a 5 m two-row four-point blade bearing with 180 mm angular contact ball bearings. The scaled bearings

were tested with 2.5 GPa and the blade bearing was tested with 2.0 GPa. He was able to produce wear for both sizes, with similar x/2b ratios (Stammler, 2020). Wandel and Bartschat compared the wear development under oscillating conditions with a low cycle number. The tested two bearing sizes (40 mm and 100 mm, both inner diameter) and stated that the development for both bearing sizes is similar (Wandel S., and Bartschat B., 2021). In addition, Wandel et al. showed that the grease parameters are a crucial factor for wear. They considered the ability of the lubricant to release oil and the mobility of the base

oil as the main parameters (Wandel et al., 2022).

Furthermore, Stammler showed that the sequence of a pitch movement plays an important role and that it influences wear occurrence. He was able to prevent wear on a scaled diameter as well as on a real size blade bearing, by adding a movement with larger amplitude to a continuous oscillation (Stammler, 2020). Table 1 summarizes the mentioned tests.



**Table 1: Overview of wear tests**

| Reference | x/2b | Frequency in Hz | Diameter in m | Cycles | Contact pressure in GPa |
|---|---|---|---|---|---|
| Becker, 2012 | unknown | unknown | (ID) 0.060 | >500,000 | 3.0 |
| Schwack, 2020 | 0.9, 13.3, and 29.1 | 2.3 and 8 | (ID) 0.040 | 250,000 | 1.9 |
| Schwack, 2021 | 2.67 and 11.44 | 0.5 | (OD) 0.750 | up to 12,500 | 2.0 |
| Wandel and Bartschat, 2021 | 1 to 31<br>2.34 | 0.2, 1, 3 and 5<br>0.5 | (ID) 0.040<br>(ID) 0.100 | 4,000<br>up to 40,000 | 1.7<br>2.5 |
| Stammler, 2020 | 2.66<br>3 | 0.5<br>0.5 | (OD) 0.180<br>(OD) 5.000 | 40,000<br>40,000 | 2.5<br>2.0 |


Blade bearings from different turbines can differ in their size, in the rolling element type, and in their contact geometry. Among others these will influence wear occurrence since they effect the oscillation parameters. Tests for any real scale blade bearing require significant funds and time. A blade bearing has typically a diameter larger than 1 m and a weight of a few tons. The handling of such components and the assembly processes require special tools and time. Furthermore, the tests are time and

energy consuming. In contrast scaled tests are cheaper and give a good understanding of principial wear mechanism, like Stammler showed (Stammler, 2020), but they cannot cover every aspect. One of these aspects is the grease and its behaviour in a bearing. The grease blend and how the distribution changes when the bearing is moving, as well as the possibility of the base oil to replenish the contact, will be different with an increased bearing diameter and contact size (FVA-Nr. 327 III, 2011). In addition, a common blade bearing steel type is 42CrMo4 (Chen et al., 2014), where smaller bearings typically made of

100Cr6 steel (Schaeffler Technologies AG & Co. KG, 2019). Therefore, full-scale tests are mandatory to understand the influence of oscillation parameters on blade bearings.

This work covers such full-scale tests and sheds light on wear in blade bearings. The tests from former works, summarized in Table 1, will be picked up and give an input for this work. The objective is to find limits for wear, to tune a blade bearing

controller to avoid these conditions, e.g., by including beneficial pitch movements. Typical wind turbine parameters of a reference turbine are used as input for tests with blade bearings with an outer diameter of 2.6 m. Section 2 explains the considered parameters. Section 3 and Section 4 describe the test rig and the test setup. The test results are presented in Section 5 and are summarized in Section 6.



## 2 Wear and Operational Parameters

The term "wear" covers different types of surface induced damages (Tallian, 1992). Wear can occur in the contacts of rolling bearings. If wear in a rolling bearing is induced by oscillating movements, possible damage modes are "stand still marks" and "false brinelling". Adhesion and tribo-corrosion are the key wear mechanisms for both. Stand still marks typically have, In contrast to to false brinelling, an undamaged inner area and show additionally a surface disruption. They are mainly caused by oscillations with a small amplitude (x/2b < 1), which can stem from vibrations even if the bearing is not turning. False brinelling

show in an early state a high amount of corrosion. The reaction products have typically a brown, red, or black colour. They are identified as Hematite ($\alpha$-$Fe_2O_3$) or Magnetite ($Fe_3O_4$) (Sommer et al., 2010). Abrasion secondly leads to a dent with a polished surface. Wear products contaminate the grease and can negatively influence its functions. Furthermore, the surface changes lead to an increased friction torque.

The rolling element squeezes the lubricant out of the contact area, if it is oscillating with a sufficiently small amplitude (Wandel et al., 2022). Therefore, it is obvious that parameters describing an oscillation should be considered. These are the amplitude, the frequency, and the velocity. The following Eq. (1) shows their correlation:

$$v(t) = y_0 \cdot \omega \cos(\omega \cdot t). \tag{1}$$

Herein, $v(t)$ is the velocity, $y_0$ the amplitude, and $\omega = 2 \cdot \pi \cdot f$ is the frequency. In addition, the number of repeated cycles

and the load influence the wear behaviour as well. It is easy to imagine that wear become worse if its trigger, the oscillation movement, is repeated for several times (Wandel S., and Bartschat B., 2021). A higher contact force results in more pronounced wear (Sommer et al., 2010). In addition, the ability of the lubricant itself to flow back to the contact area, the base oil viscosity, and the temperature, are important as well (Wandel et al., 2022). But they are not part of this investigation since all bearings were tested with the same grease and with comparable temperature conditions.

## 3 Test Rig

The structure and stiffness of the hub and the blade influence the load distribution (Stammler et al., 2018b). In combination with the applied forces and moments, they cause a complex load distribution in the bearing. Hence, the wear characteristic would be different on every position around the circumference. This can cause interaction between the differently loaded rolling bodies and makes it difficult to determine the influence of individual oscillation parameters. Therefore, a test rig that

applies a uniform load appears to be suitable and has the advantage of being cost efficient due to a simpler design. Although the load distribution is not realistic, the conclusions can be transferred since the most highly loaded rolling elements have a similar load. The weakly loaded rolling elements would have a less severe characteristic or would not produce wear.
The so called BEAT2.2 (Bearing Endurance and Acceptance Test rig) tests two standard serial bearings with an outer diameter of 2.6 m. The bearing type is a two-row four-point contact ball bearing with 196 balls. The bearings are inductive hardened





and have hard-turned raceways. Prior to the assembly and filling with grease, the rings were cleaned by mechanical means. A commercial grease with a base oil viscosity at 40°C of 50 cSt, a NLGI class of 2, and a Lithium thickener is used. Table 2 lists further dimensions of the test bearing.

**Table 2: Bearing Dimensions**

| Property | Value | Unit |
|---|---|---|
| Inner diameter | 2070 | mm |
| Outer diameter | 2600 | mm |
| Pitch diameter | 2310 | mm |
| Number of rows | 2 | - |
| Rolling elements per row | 98 | - |
| Rolling element diameter | 65 | - |
| Initial contact angle | 40 | ° |
| Inner and outer osculation | 0.53 | |


Figure 1 shows the test rig. An intermediate ring (4) connects the rotating outer rings of the upper (1) and lower bearing (2). The intermediate ring gives enough space for a hydraulic system (3), which applies the load. It is placed between the upper and lower inner ring and consists of 50 cylinders that apply a static load up to 10 MN. An electric pitch drive controls the pitch movement with a top speed of 6 °/s. A torque measurement is mounted to the pinion shaft.



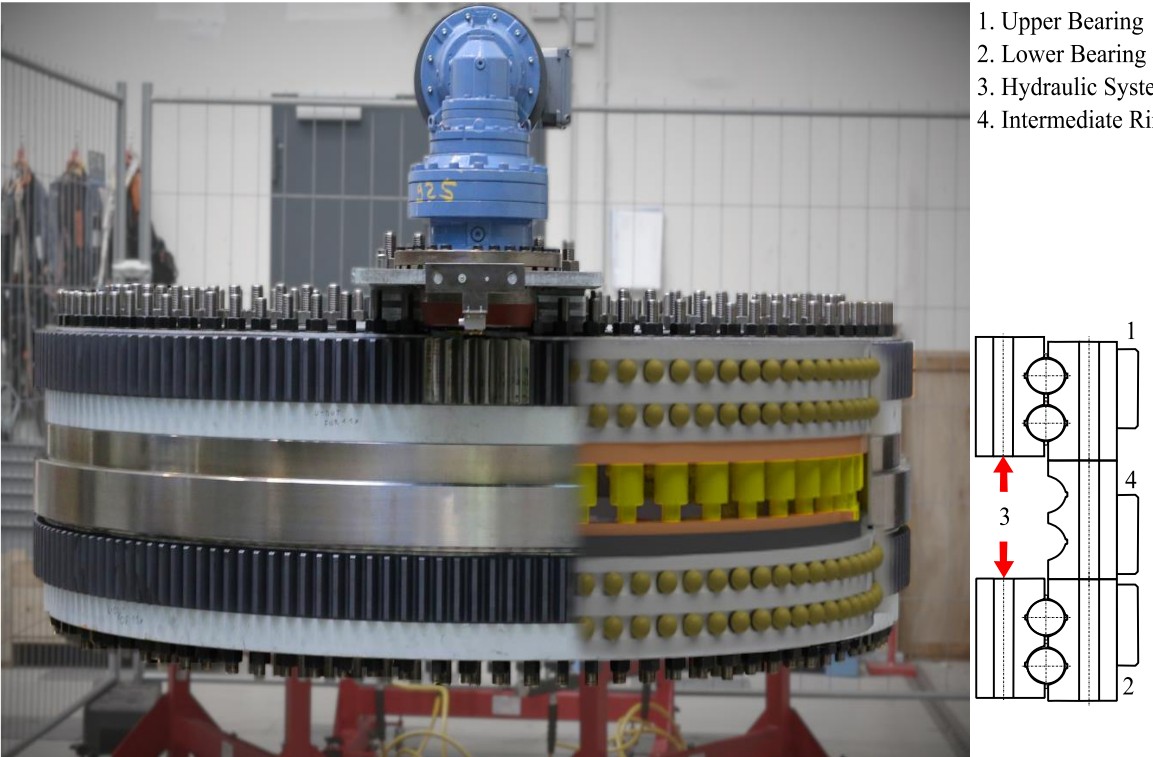

1. Upper Bearing
2. Lower Bearing
3. Hydraulic System
4. Intermediate Ring

**Figure 1: Test rig BEAT2.2 (photo by Karsten Behnke, visualisation by Filip Kovacevic)**

The applied axial load must be transferred to a value which allows to evaluate test results. Hence, it must correspond to the rolling elements. The rolling element load can be calculated from the acting load and allows to determine the Hertzian parameters, like the contact pressure. Among others, Houpert published a method to calculate the Hertzian parameters starting with the individual rolling element loads (Houpert, 2001). The contact pressure can also be used to compare test results to each other.

An estimation of the individual rolling element loads is possible neglecting elastic behaviour. A more detailed look requires a finite element (FE) calculation, where elastic ring deformation and other effects can be considered. As a detailed modelling of the contacts between rolling bodies and the raceways would lead to a high computational effort, it is a common approach to represent the rolling bodies with nonlinear spring elements. Dadié published an approach based on nonlinear traction springs which connect to the raceways by rigid beam elements (Daidié et al., 2008). Following this approach, a global FE model of the tested bearings is developed via APDL (ANSYS Parametric Design Language) in ANSYS Classic (Schleich, 2019). Based on this global FE bearing model, a full test rig FE model is developed. Figure 2 shows a cross sectional view of the test rig. The outer ring and inner ring bolts are modelled with beam elements and friction-based contacts are defined between the model components. LINK11 elements are used to model the hydraulic actuators between the inner rings of the test bearings. These elements connect to the surrounding components by constraint equations (RBE3) that equally distribute the load on the master



node to the slave nodes. By means of this model it is possible to analyse the resulting contact forces with considering elastic behaviour.

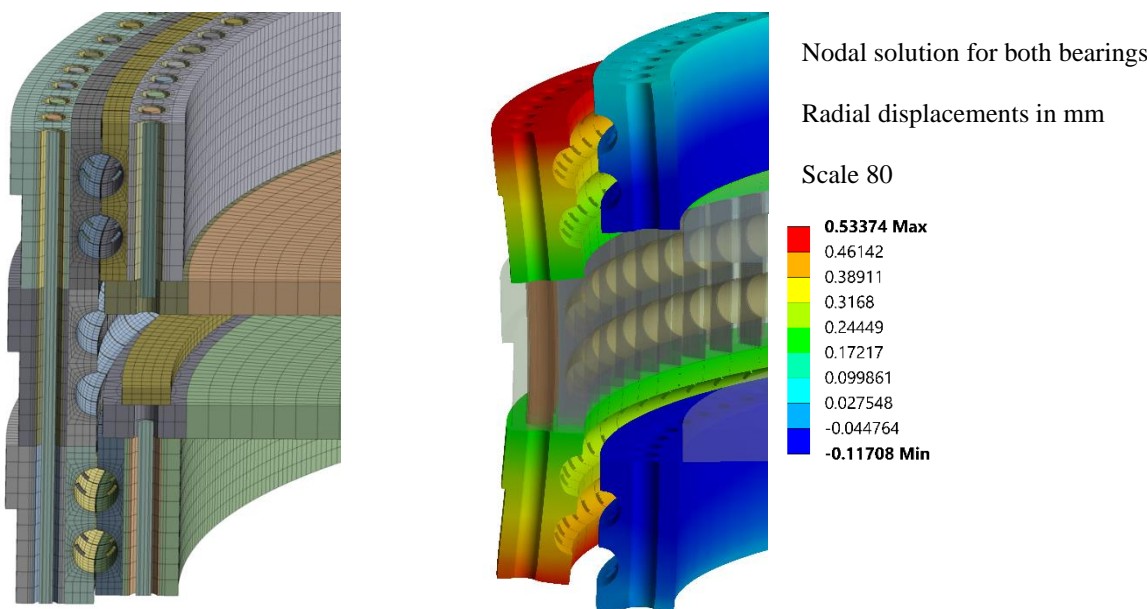

Nodal solution for both bearings

Radial displacements in mm

Scale 80

| | |
|---|---|
| 0.53374 Max | |
| 0.46142 | |
| 0.38911 | |
| 0.3168 | |
| 0.24449 | |
| 0.17217 | |
| 0.099861 | |
| 0.027548 | |
| -0.044764 | |
| -0.11708 Min | |

**Figure 2: Cross sectional view of the developed FE test rig model in undeformed (left) and deformed state (right)**

Simulations are performed for different load levels. For each load level all implemented hydraulic actuators apply the same force to the structure. Figure 3 shows the load distribution on the raceways of both test bearings for an exemplary axial load of 8.0 MN. While in both bearings the resulting load distribution is nearly identical, a significant difference for the contact forces on the different rows of each bearing can be seen.


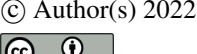



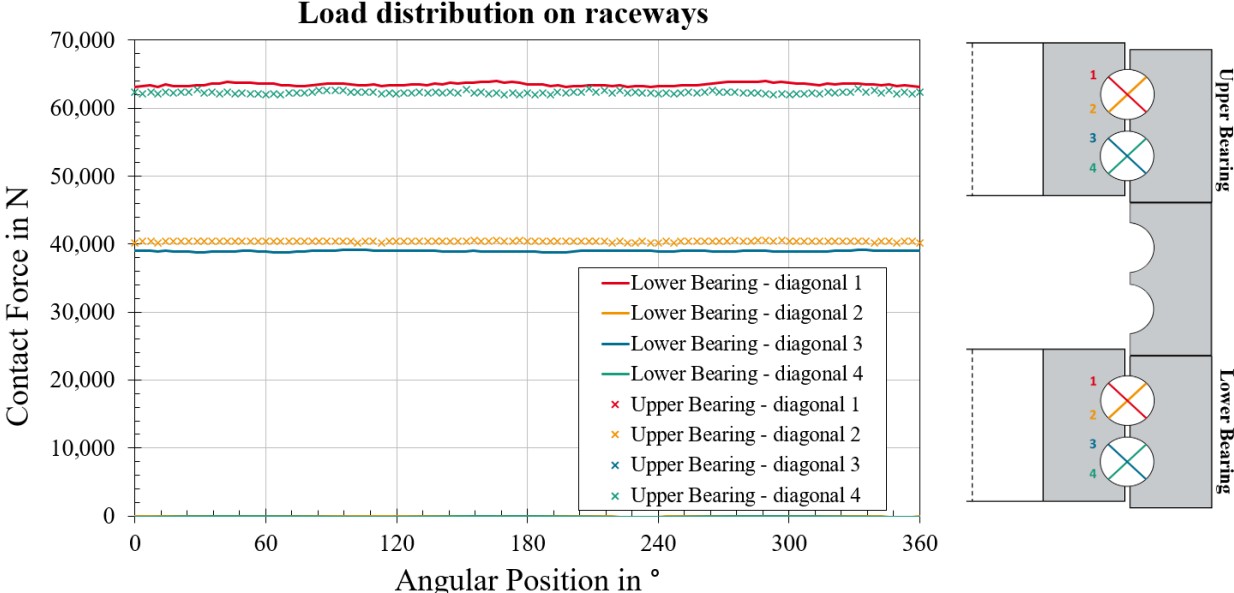

**Figure 3: Resulting load distribution on the raceways of both test bearings for 8.0 MN total axial force**

This difference is caused by the elasticity of the bearing rings. Furthermore, it shows that just one of the two diagonals of each rolling element transmits loads. With these simulations it is possible to ensure that the desired contact pressure between rolling element and raceway is achieved for the tests.

**4 Data Bases**

The range for the test parameters is based on two inputs. On the one hand, they should fit and complement the published tests (see Table 1) and, on the other hand, they should cover realistic turbine operation conditions. The analysis of aeroelastic simulations of a reference turbine ensures the latter. The reference turbine is designed for wind class IA according to International Electrotechnical Commission (IEC) 61400 (IEC, 2019), has a rated power of 3 MW, a hub height of 85 m, and a rotor diameter of 100 m (Leupold et al., 2021). While the test bearings are not specifically designed for this reference turbine, their dimension can be found in similar commercial turbine designs. The analysis is limited to design load case (DLC) 1.2 which covers the normal power production of the turbine.

Figure 4 shows the cycle count for a CPC and IPC controller of this turbine. One bar indicates the number of cycles within 20 years in a double amplitude span of 0.5°. For the CPC, most common blade angle adjustments have less than 0.5°. If IPC is active, the maximum is shifted towards movements with a few degrees. Oscillations with a double amplitude higher than 5° are unlikely, especially for the CPC controller. Hence, the test range covers double amplitudes up to 5°.

The second test parameter is the load. The test rig applies an axial force, whereas a blade bearing is loaded by bending moments, axial forces, and radial forces. In terms of wear tests, the contact pressure in the rolling contact is sufficient to transfer one to





the other load condition. The rolling element load, which is necessary to calculate the contact pressure, can be calculated with an FE or with a simplified empirical approach. The second neglects elastic behaviour and gives less accurate results.

Figure 5 introduces the classification of the reference turbine blade root bending moment. It considers a lifetime of 20 years. The resulting bending moment acting at the blade root has the major influence on the bearing loads (Stammler et al., 2018b). A frequently resulting moment is 2.5 MNm. With both mentioned simplifications, it leads to the highest contact pressure of

approximately 2.0 GPa. In the same manner, 5.5 MNm results in a contact pressure of 2.5 GPa. The range from 2.0 to 2.5 GPa represents the average values of published research in this field (see Table 1) and defines the test loads. The load case, shown in Figure 3, corresponds to 2.5 GPa contact pressure at the higher loaded row.

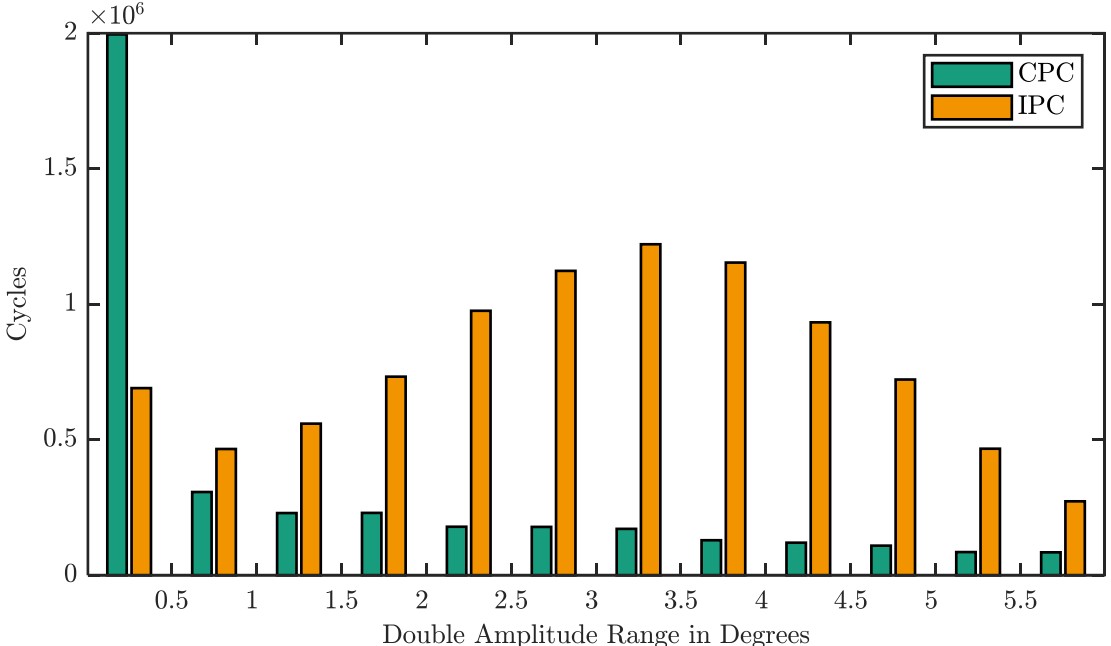

**Figure 4: Cycle comparison of CPC and IPC**



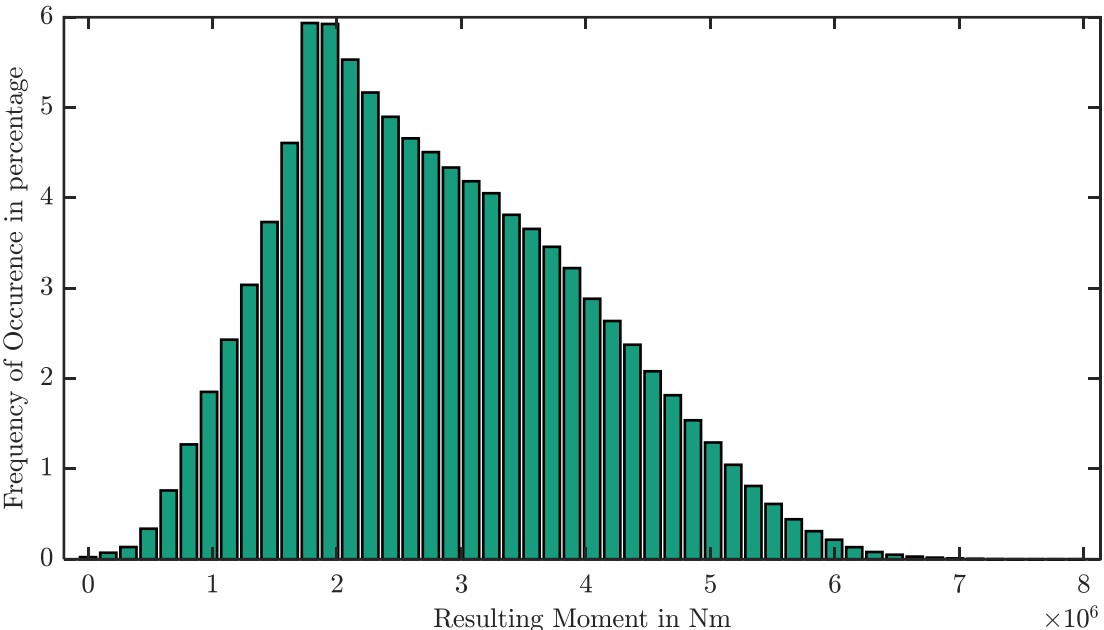

**Figure 5: Classification of the resulting moment**

Beside the load and amplitude, the velocity and frequency complete the tests setup. The typical pitch speed of the reference turbine is less than 1°/s. The tests comply to this requirement. One additional test has a higher speed, which matches to the maximum pitch speed. The speed varies in a sinusoidal profile. The given speed corresponds to the maximum speed in a cycle. The frequency follows as given in Eq. (1). The frequencies and hence the speed from the literature in Table 1 have the tendency to be higher. It could be done to shorten test time or to keep a similar entrainment speed as for a real blade bearing.

According to Wandel and Bartschat even a few thousand cycles lead to wear (Wandel S., and Bartschat B., 2021). The tests in this work have cycles numbers in the range of 1,000 to 40,000. Table 3 summarizes the test parameters. The contact pressure refers to the most highly loaded row.

**Table 3: Test setup**

| Test ID | x/2b | Amplitude in ° | Contact pressure in GPa | Maximum speed in °/s | Frequency in Hz | Entrainment speed in mm/s | Number of cycles |
|---|---|---|---|---|---|---|---|
| I | 2.9 | 0.40 | 2.55 | 0.8 | 0.32 | 7.72 | 1,000; 3,000; 5,000 |
| II | 3.6 | 0.40 | 2.05 | 0.8 | 0.32 | 7.72 | 1,000; 3,000; 5,000; 20,000; 40,000 |
| III | 18.4 | 2.50 | 2.55 | 5.0 | 0.32 | 48.25 | 40,000 |





| IV | 18.4 | 2.50 | 2.55 | 0.8 | 0.05 | 7.72 | 40,000 |
| V | 8.1 | 1.00 | 2.24 | 0.8 | 0.13 | 7.72 | 5,000 |
| VI | 1.0 | 0.15 | 2.55 | 0.3 | 0.32 | 2.89 | 5,000 |

## 5 Test Results and Discussion

### 5.1 Tests with Steady Oscillations

As described in Section 2 it is expected that wear damage becomes worse with increased load and more cycles. The test results of Test ID I and II support this assumption. Figure 6 shows wear marks with an increased cycle number and with two different loads. The three tests on the left side were performed with 2.5 GPa contact pressure (Test ID I). The cycle number is increased from left to right (1k, 3k, 5k cycles). The other five have a lower contact pressure, they were tested with 2.0 GPa (Test ID II). The cycle number grows as well from the left to the right (1k, 3k, 5k, 20k, 40k cycles). The maximum velocity ($v_{max} =$

$0.8\,°/s$), the amplitude ($y_0 = 0.4°$), as well as the frequency ($f = 0.32\,Hz$) were kept the same for all these tests. The wear marks look similar to the published ones mentioned in Section 1. Especially the results from Wandel and Bartschat, who tested smaller bearings with a comparable number of cycles (Wandel S., and Bartschat B., 2021).

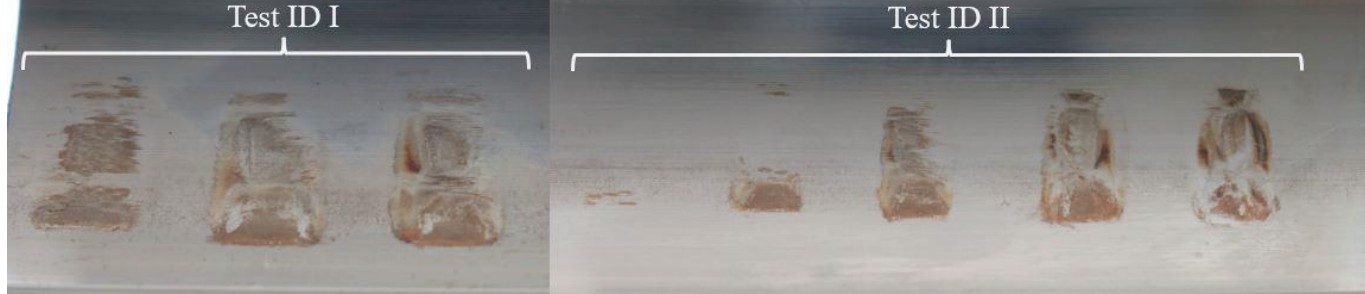

**Figure 6: Three wear marks obtained with contact pressures of 2.5 GPa in test ID I (left), five wear marks obtained with contact pressures of 2.0 GPa Test ID II (right)**

It is possible to see that the damage is visually more pronounced at higher loads and higher cycles. The width of the wear area at higher loads is slightly larger due to the increased hertzian contact width (2b).

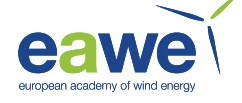
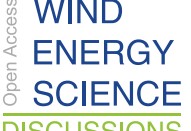

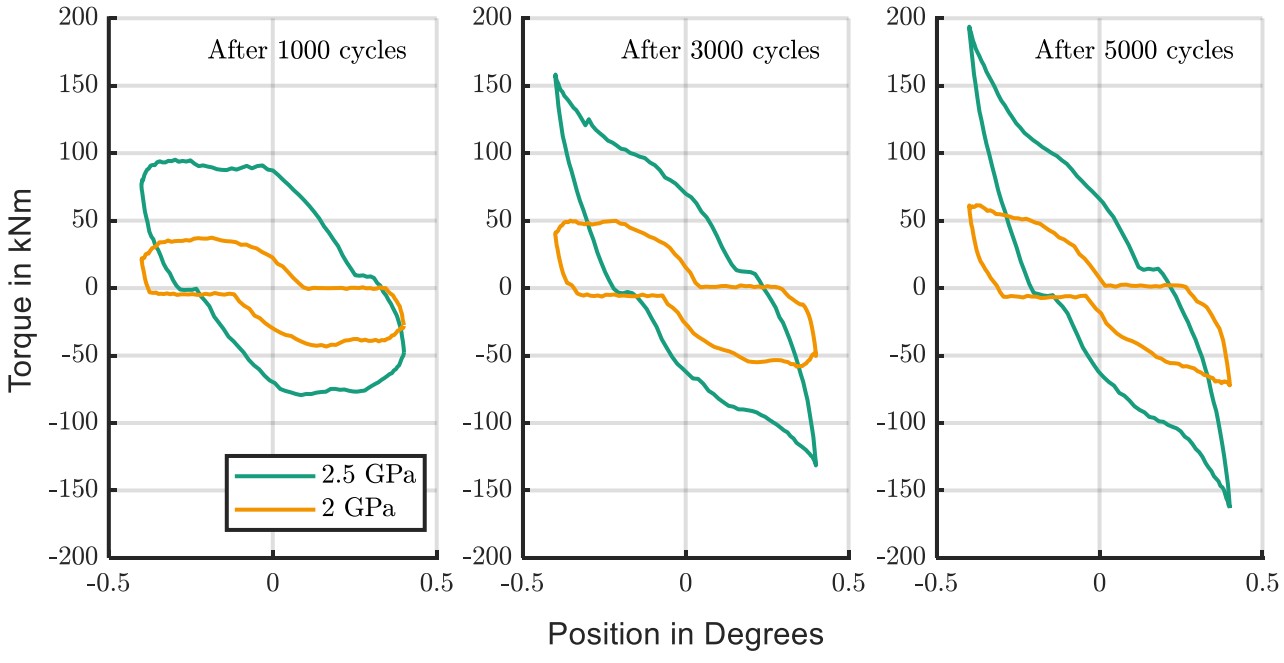

**Figure 7: Torque development, comparison of contact pressures of 2.5 GPa (ID I) and 2.0 GPa (ID II)**

Furthermore, it is possible to see that fretting corrosion occurs even at low cycles. The friction torque confirms this visual impression. Figure 7 shows the measured torque of both load steps for one cycle, after 1k, 3k, and 5k repetitions. The torque rises with more cycles, and it rises faster for the higher load. It confirms Bartschat's and Wandel's conclusion that under certain circumstances wear develops quite fast (Wandel S., and Bartschat B., 2021). The correlation between the severity of the wear mark and the cycle number or the load is quite simple to summarize: With an increased cycle number the wear damage becomes worse.

When it comes to the influence of amplitude, speed, and frequency it is not as simple. Eq. (1) gives the relation for these three parameters. When one of it changes, at least one of the other two will change as well. These circumstances lead theoretically to a high number of necessary tests to analyse the influence of every parameter individually. The tests are reduced to a few by using parameter combinations that are relevant for the turbine operation conditions. At first it is worth to look at the speed and frequency. Figure 8 introduces Test ID III and IV, which have the same amplitude ($y_0 = 2.5°$), but a different velocity respectively a different frequency ($f_{left} = 0.32\ Hz, f_{right} = 0.05 Hz$). They compare the influence of the speed and frequency. The wear mark of the faster test speed shows a higher corrosion.





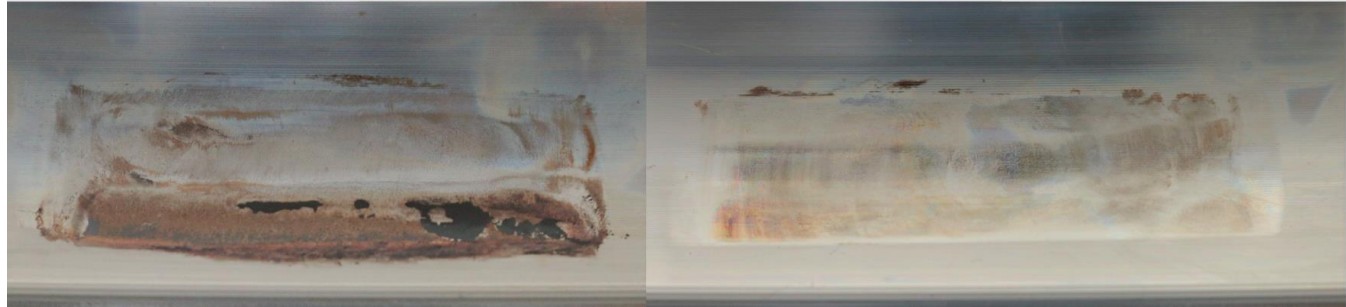

**Figure 8: Two wear marks with 2.5° amplitude; left: tested with 5°/s (ID III), right: tested with 0.8°/s (ID IV)**

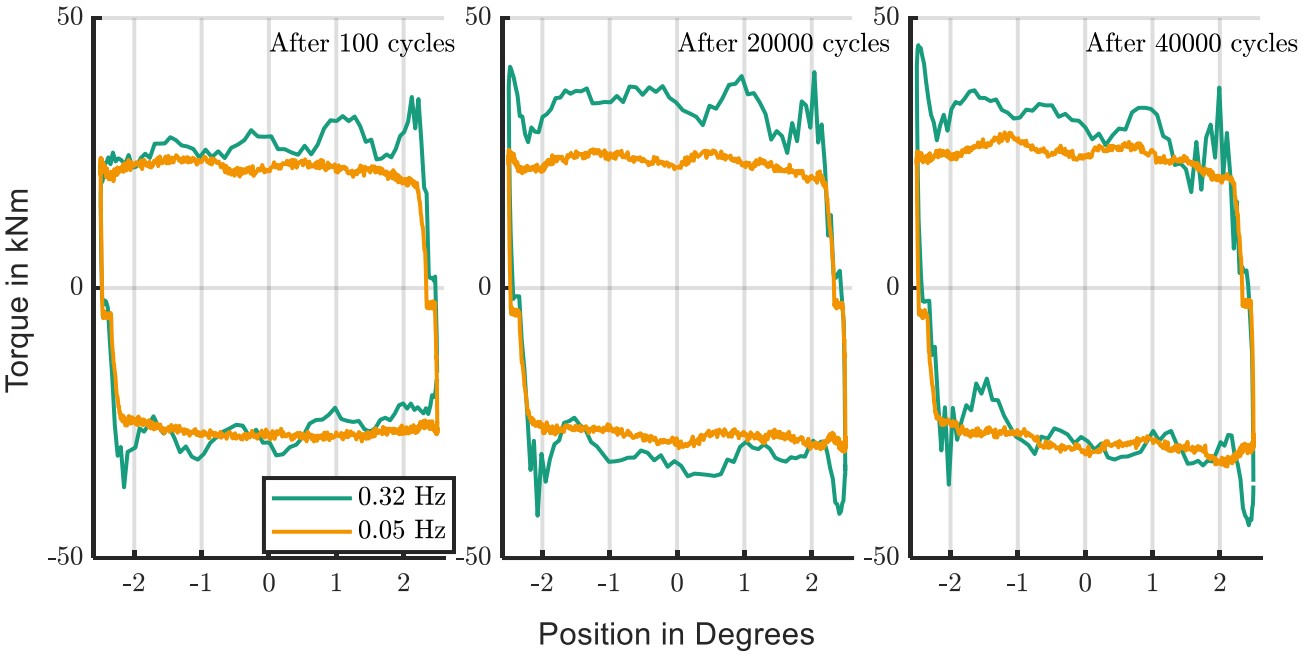

**Figure 9: Torque development with increasing number of cycles, comparison of oscillation frequencies of 0.32 Hz and 0.05 Hz**

Figure 9 shows the friction torque of one cycle at the beginning, in the middle and at the end of the tests. The initial torque of the fast test is slightly higher, due to a higher inertia force with higher velocity and acceleration. It also increases more strongly during the test. It is possible to conclude that speed and frequency influence wear. This hypothesis is supported by Wandel (Wandel et al., 2022), who states that with a lower frequency there is more time for the grease to relubricate the contact. Furthermore, a rebuild of a protective layer takes time, hence oscillations with a lower speed have a lower risk for wear (FVA-Nr. 327 III, 2011).

The first three tests (Test ID I) in Figure 6 have an amplitude of 0.4° and were done with the same contact pressure. The left test in Figure 9 (Test ID III) has the same frequency, where the right one (Test ID IV) has the same maximum speed. The




visual impression states that the damage marks with the same frequency look more alike. This supports the earlier assumption about the importance of time for contact replenishment and tribo-layer build-up.

Test ID I and III also allow to examine the influence of the amplitude. A significant rise in torque could be obtained faster for tests with a smaller amplitude. The test with a small amplitude (ID I) led to a seven times higher peak torque after 3k cycles

(Figure 7). The torque development of the test with a higher amplitude (ID III) stays below that value. It was nearly doubled after 40k cycles. In other words, smaller amplitudes seem to influence the behaviour of the bearing quicker. In addition, it is worth to mention, that for the high amplitudes of 2.5° the different wear marks have a less uniform appearance. A significant amount of them have less severe characteristics. It supports the statement that larger amplitudes are less critical. It was also confirmed for high x/2b ratios by Schwack et al., that have seen similar (Schwack et al., 2020).

Test ID V and VI complement the results in terms of the amplitudes influence. The right side of Figure 10 presents a test with an amplitude of 1° (ID V) and on the left side a test with a small amplitude of 0.15° (ID VI). In Figure 11 compares the torque development of these tests to the previous ones. Test ID V fits perfectly and supports the statement, that wear severity decreases with higher amplitudes. The test with the smallest amplitude diverges. It has a slightly lower characteristic.

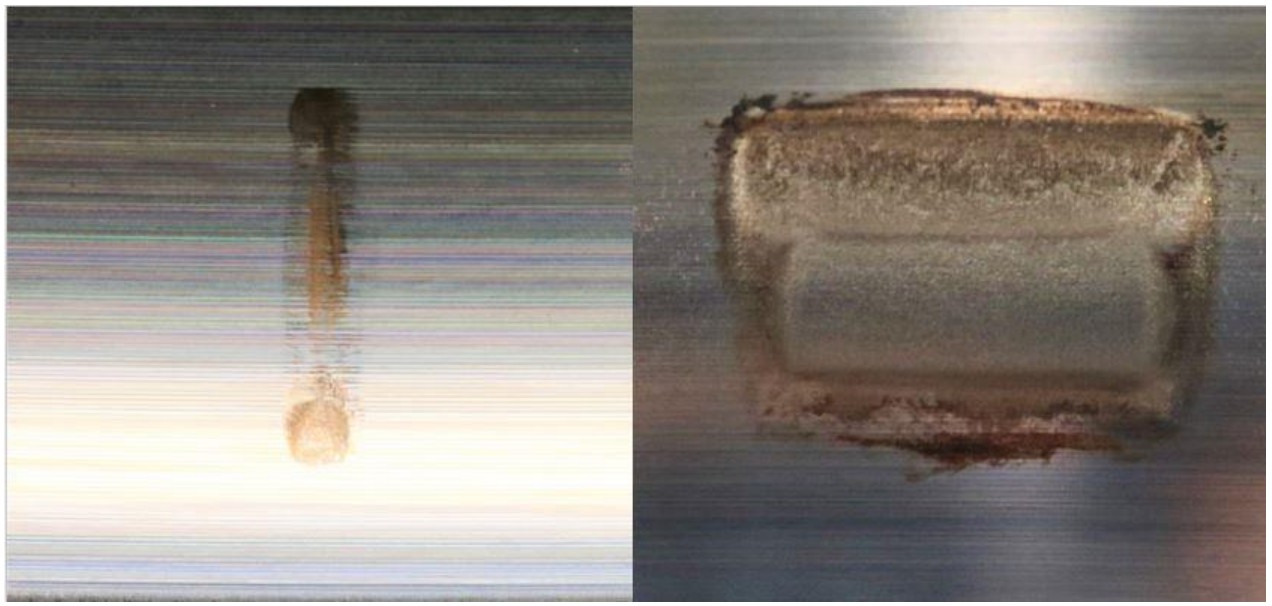

**Figure 10: Two wear marks with different sizes; left: 0.15° test amplitude (ID VI), right: 1° test amplitude (ID V)**





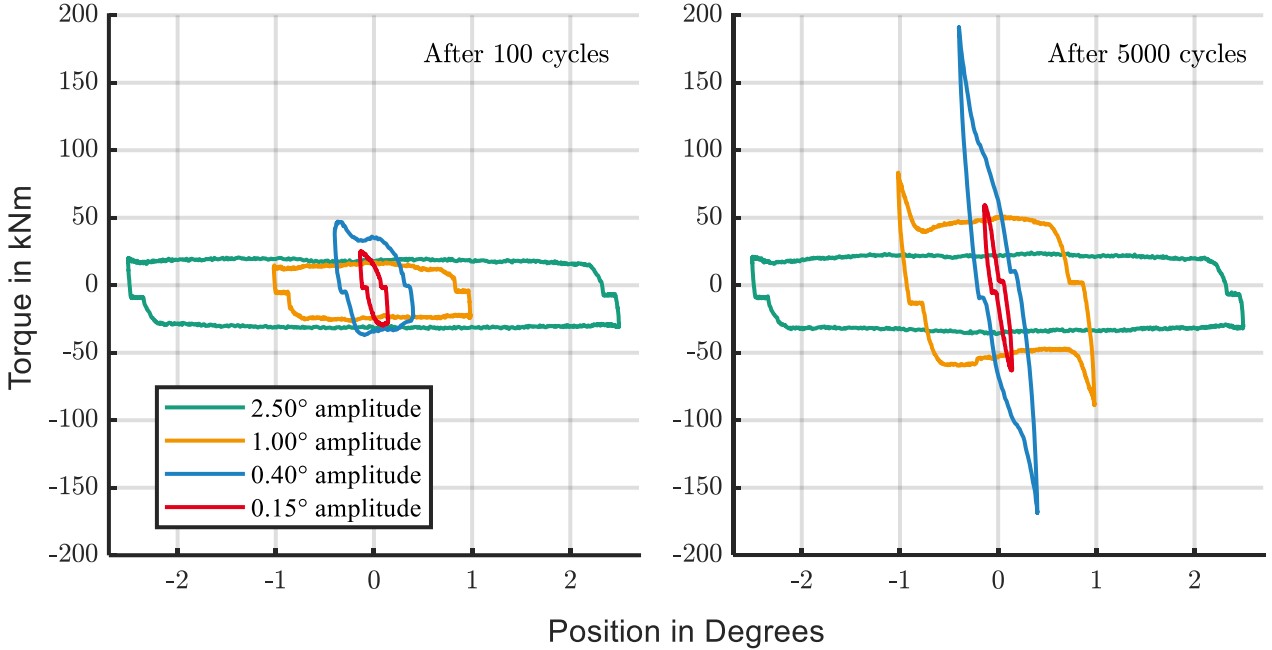

**Figure 11: Torque development, comparison of different amplitudes (ID I, III, V, VI)**

The similarity in x/2b ratios allows to compare the damage marks off Test ID I (x/2b = 2.9) with the published results of Schwack (Schwack, 2020) and Stammler (Stammler, 2020), see Table 1. Figure 12 depicts wear marks for bearings ranging from 80 mm to 5000 mm outer diameter. From left to right it starts with a type 7208 bearing with 80 mm outer diameter and a bearing with 750 mm outer diameter (Schwack, 2020). Followed by Test ID I, with 2.6m outer diameter and finally a 5000 mm bearing (Stammler, 2020). While different kinematics off the contacts results in different shapes of the damages, the primary damage product hematite and the primary mechanism fretting corrosion are the same over the entire range.






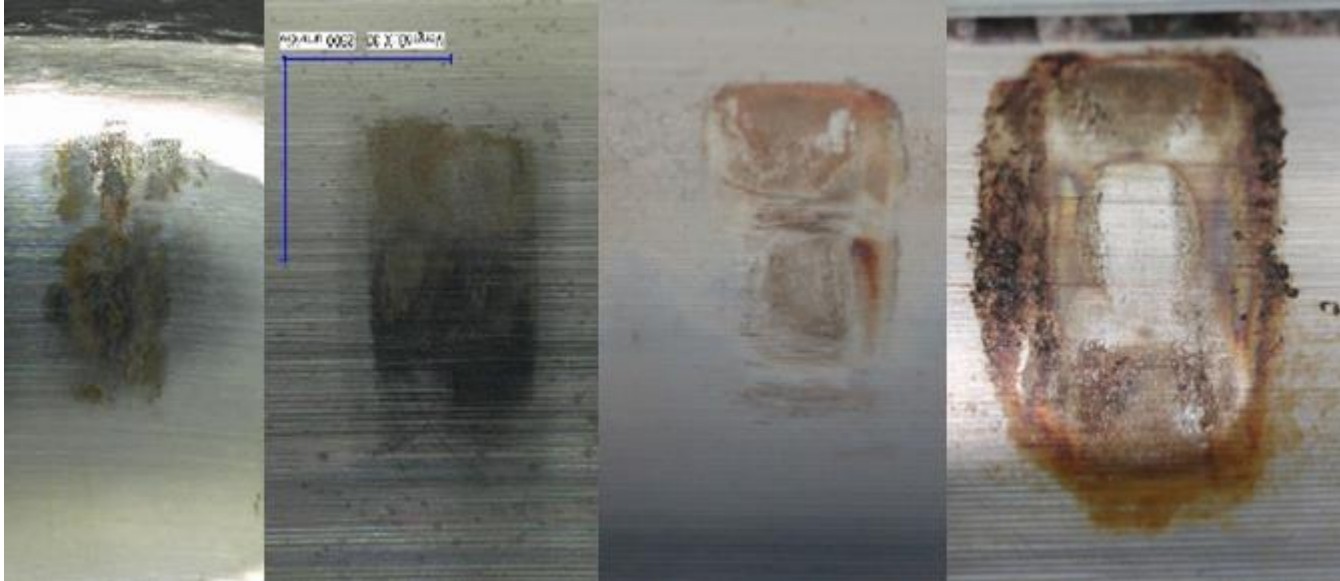


**Figure 12: Wear with similar x/2b on bearings ranging from 80 mm to 5000 mm outer diameter (Schwack, 2020; Stammler, 2020)**

**5.2 Tests with Protection Runs**

All tests described so far were carried out without interruptions. An interruption could be a movement with a different
amplitude, which will be called protection run. Stammler first demonstrated the ability of longer movements to prevent wear
when embedded in oscillation with a small amplitude which on their own would cause wear (Stammler, 2020).

Tests with one longer cycle after a particular number of the base oscillations were done. To compare the results with a
continuous oscillation as reference Test ID I is used. The protection runs have a tenfold higher amplitude (4°) than the base
oscillations (0.4°) and seem to be effective to avoid wear. If the protection run occurs every 10 or every 50 cycles, it is not
possible to observe wear on the raceway after a repetition of 40k cycles.

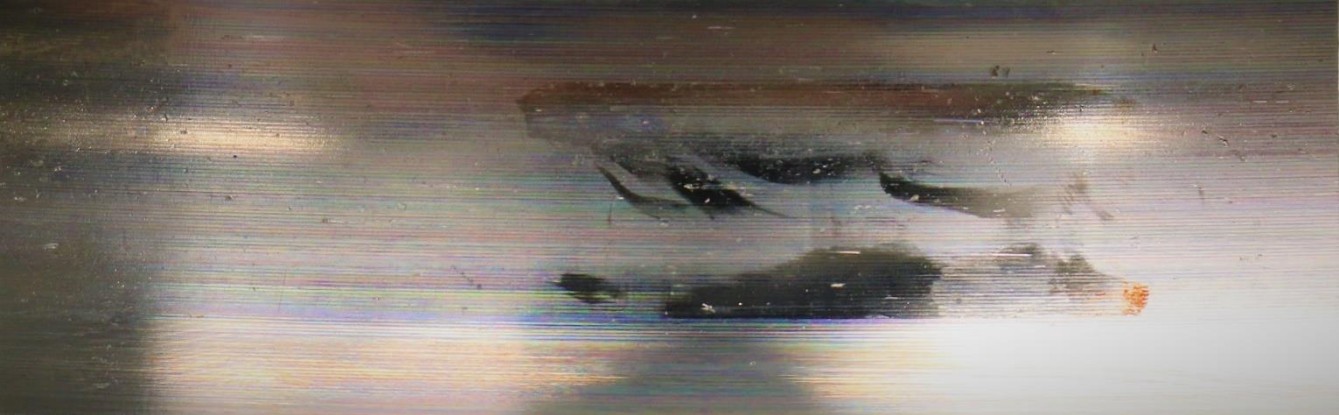

**Figure 13: Test with protection run every 100 cycles**




If the protection run frequency is reduced to every 100 cycles, there is beginning wear mark visible, compare Figure 13. Nevertheless, this mark looks tremendously better than every other shown in Figure 1. Fretting, like in the bottom right corner,
just occurred for some of the rolling elements. A polished raceway surface indicates the ball movement during the protection run. There is no significant change in the torque for all these three protections runs. In Figure 14 the curve of the test with a protection run every 100 cycles shows a slightly increased torque, which fits to the visual impression. However, the torque stays below 30 kNm, which is lower than for the reference test, shown in Figure 7.

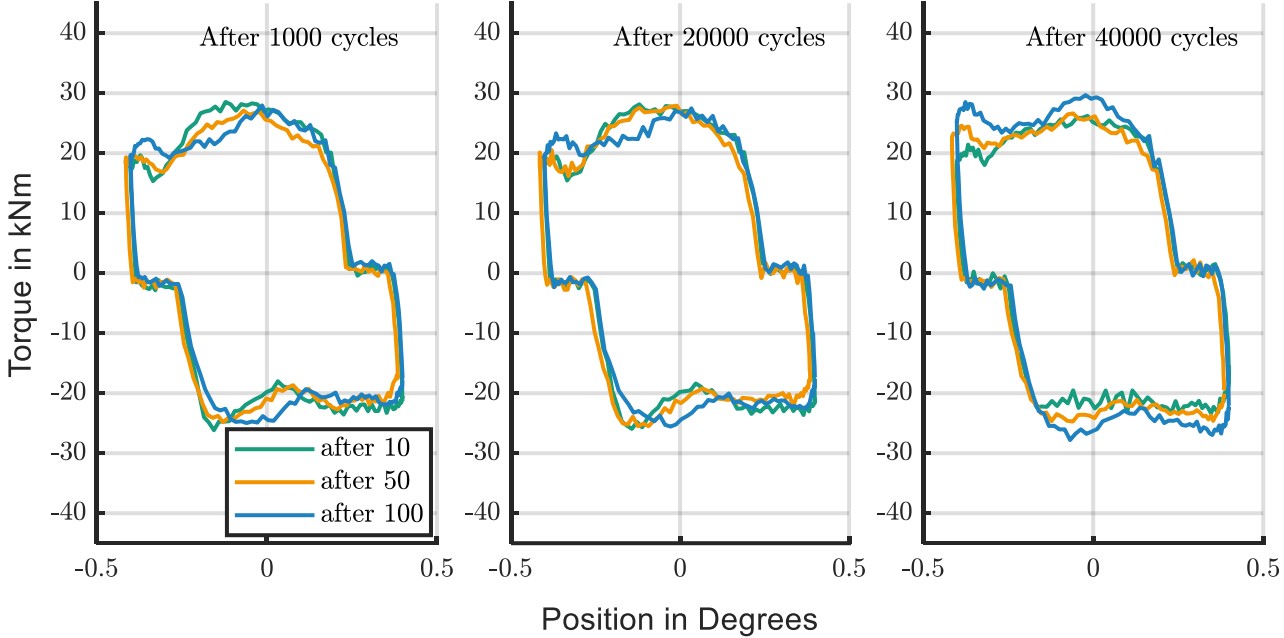

**Figure 14: Torque development, comparison of protection runs with different frequencies of occurrence**

The next test has a shorter protection run. Its amplitude (0.8°) is twofold the regular amplitude. It was executed every 50 cycles. The test was performed with 2.0 GPa, the reference test is Test ID II.



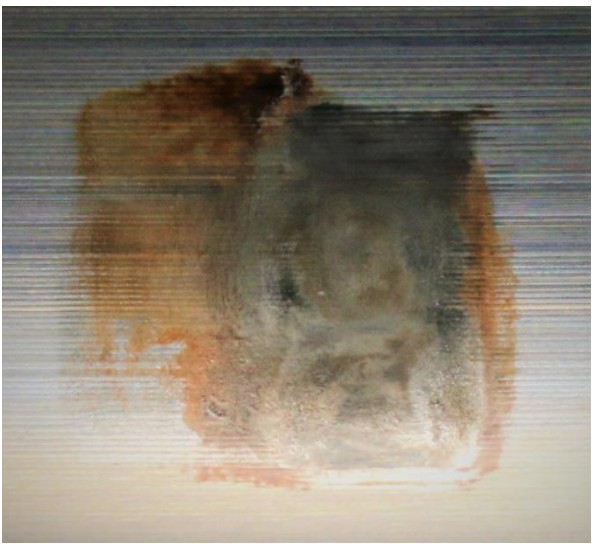


**Figure 15: Wear mark from a test with failed protection run**

In contrast to the other three tests with a protection run of 4°, this test, with a smaller protection run amplitude, was not able
to prevent wear. Figure 15 shows the wear mark, which has two areas: one area based one the small cycle (right), the other
half covering the area where the ball moved during the protection run (left). Corrosion occurs in the entire area. Figure 16

compares the friction torque to the reference test. The torque of the reference test has a similar shape, but a higher peak.
Nevertheless, the friction torque shows also that this protection run was not effective. Compared to the torque of the other
three (Figure 14), which were tested with a higher load, the torque is higher, and it rises over time.





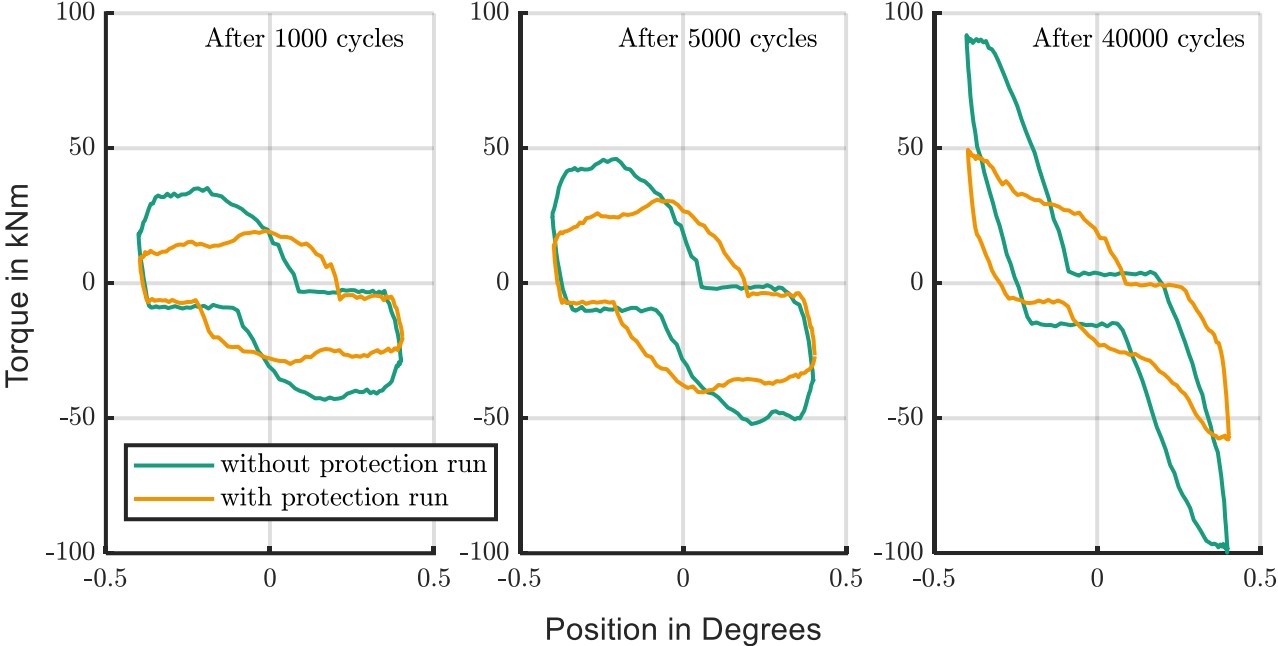

**Figure 16: Torque development, comparison of protection run and reference test**

## 6 Conclusions and Outlook

This work covers the test of blade bearings with an outer diameter of 2.6 m on a sandwich-type test rig. The test parameters are based on a 3 MW reference turbine and are compared to values from the literature. The bearings were tested under oscillating conditions with a steady axial load. The x/2b ratios range from 1 to 18.4 with peak speeds between 0.3 °/s and 5 °/s. All wear marks show reddish, brown oxide layers. Hence one of the wear mechanisms is fretting corrosion. Every tested parameter combination of load, amplitude, frequency, and speed with a steady oscillation shows wear and lead to a rise in torque. That means, every considered operation condition can potentially lead to wear in a blade bearing. It is possible to define parameter combinations that are more or less likely to promote wear. The test results confirm that wear becomes severer with higher loads and higher oscillation speeds. Furthermore, wear increases with the number of cycles the bearing is exposed to. The influence of the amplitude is not as clear as the influence of load, speed, frequency, and number of cycles. The severity of wear has a maximum in the middle range of tested amplitudes at 1° (cf. Table 3). Cycles with higher or lower amplitudes are less harmful.

The initial motivation of the tests was to find thresholds for wear occurrence within typical operational parameters of wind turbines. With steady oscillations and loads, however, wear cannot be prevented. On the other side, the effectiveness of protection runs could be confirmed with this bearing size. Incorporating this concept, it is possible to define a limit for wear.



This limit needs to include the number of cycles between protection runs and their parameters, since not every protection run was effective.

Bartschat and Schwack showed that the application of additional protection runs can have a negative effect on the turbine operation, since a changing pitch angle will influence the turbine loads as well as the energy output (Schwack et al., 2018). Therefore, the objective of further investigations is to find a minimum frequency of protection run executions and to find a

minimum effective amplitude.

The grease and its ability to lubricate a contact under oscillating conditions influences wear occurrences. Tests with different greases should also be included in further work.

**Author contributions.** KB designed the experiments, carried them out and did the evaluation of the test results. FS prepared

and carried out all FE simulations.

**Competing interests.** The authors declare that they have no conflict of interest.

**Acknowledgements.** This research has been supported by the German Federal Ministry for Economic Affairs and Climate

Action with the project "HBDV - Auslegung hochbelasteter Drehverbindungen" (grant no. 0324303D). The project funding is kindly acknowledged.





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
