# Peer review of "Exploring Limiting Factors of Wear in Pitch Bearings of Wind Turbines with Real Scale Tests"

_Wind Energy Science, 2022_

## Referee Comment (RC1)

**Review**

**General comments**

In my opinion, the publication represents a significant contribution to scientific progress in the context of WES. It is of interest to the entire wind power community. It should be emphasized that the investigations were carried out on large bearings how they are used in blade bearings. The majority of previous investigations on this topic took place with significantly smaller bearings. With such laboratory tests, the question of scalability and transferability always arises.

The scientific quality of the work is good. The experiments carried out are well described and evaluated. It is clearly described why which test parameters were chosen and what assumptions were made. In addition, there is the computer simulation to check the specifications. In addition to the wear, the torque measurement over the test time also provides important information on the degree of damage.

I would like to make one small restriction regarding the references: There are many very recent literature references from the last two years. However, these are often based on findings that were published much earlier (remarks see "Specific Comments").

The structural, linguistic and graphic quality of the publication is very good. The work is clearly structured and the tables, graphs and pictures are easily recognizable and informative.

**Specific comments**

In line 11 it says "The size of the bearings and the test parameters differ from other published test results for oscillating bearings." In this case, it should already be added in the abstract what the concrete differences are.

In my opinion, the statement that damage always occurs must be put into perspective (line 14). Only one lubricant was examined. Work by C. Schadow, S. Tetora and M. Grebe, for example, clearly shows the influence of the lubricant on the development of damage.

x is defined as the complete travel path of the roller within the raceway (the term "half cycle" can be confusing) (line 43).

In line 71, a few more references should be added, e.g. from M. Grebe, C. Schadow or S. Tetora, who have already made these statements much earlier.

In line 95, the publication by Grebe from 2006 or 2008 should be listed, who already describes these effects in this way.
*Reference:*
*M. Grebe and P. Feinle:. Brinelling, False-Brinelling, false False-Brinelling; Annual Meeting of the German Tribology Society (GfT);. Proceedings, pages 49/1–49/11, Aachen, 2006.*
*First English-language mention 2008:*
*Grebe, M.; Feinle, P., Hunsicker, W.: Various Influence Factors on the Development of Standstill Marks (False-Brinelling Effect); DVM Meeting, Aachen, 2008*

In line 103, the work of C. Schadow and S. Tetora from Magdeburg should be mentioned.

In Table 2, the roughness of the treads would be an interesting parameter.

In line 129, the size of the contact surface should be added as another important parameter that can be calculated using Hertzian formulae.

Line 189 and 214: The fact that damage can occur after only a few cycles was first published in
*Grebe, Feinle, Hunsicker: Various Influence Factors on the Development of Standstill Marks (False-Brinelling Effect), ELGI-Meeting Lisbon, 2008.*
Likewise, the influence of the frequency (line 231).

In line 234, 246 and 305, the corresponding x/2b ratio should be given in addition to the angle.

**Technical corrections**
none

---

## Referee Comment (RC2)

**Review**

**General comment:**
The paper makes a significant contribution to scientific progress in analysis and data regard to WES scope. Main objective of the paper is to find limits of operating parameters in the pitch bearing with regards to wear which is to some extent new concept and has not studied extensively.

The paper did a study on the pitching operating parameter with consideration of sufficient cases. The test case is a bearing that is used in multi-megawatt wind turbine which is beneficial to apply in large wind turbine. The experiment set-up, test rig and the procedure are described thoroughly. Moreover, the results are presented quite in-depth.

There are some general notes as following:
- The relation between wear and torque is not established sufficiently.
- Some related previous work such as Yang et al.[1] considered wear volume as a wear index. It would be beneficial to express the priority of the wear index in this study compared to previous work such as Yang et. al.
- The paper doesn't clearly specify the wear marks happen in which rings (inner or outer) and the reason and discussion on it.
- Mesh independency of the finite element results are not stated.
- In the finite element modeling, tolerances and internal dimensions between the bearing rows are not considered. It is recommended to put some notes about the reason of not considering in the respective section.

**Specific comment:**
- Most frequent resulting moment according to Figure 5 is less than 2.0 MNm. Clarified the sentence "**A frequently resulting moment is 2.5 MNm**" in line 174.
- In line 88 there are repeated "**to**", and one should be cleared.
* * *
[1] Feng Yang, Huang Xiaodiao, Chen Jie,Wang Hua, Hong Rongjing "Reliability-based residual life prediction of large-size low-speed slewing bearings", Mechanism and Machine Theory 81 (2014) 94–106

---

## Referee Comment (RC3)

**Review**

**General Comments**

The authors should be commended for the work displayed in this manuscript. The relevance to the field of wind energy is substantial. First, given the scale of the bearings used in the study and the due diligence undertaken to match the operating conditions of a reference turbine. Second, the particular focus towards discussing the results by tracing back to previous research conducted on smaller bearings is useful and practical contribution to the field. And ultimately, the level of openness in clarifying the methodology, and experimental details involved. Furthermore, the presentation of the results and the images is clear, and the text is of good quality. Some minor suggestions are listed below under specific comments.

**Specific comments**

- One of the main findings stated in the abstract is the non-existence of "wear limits". This statement should be conditioned, at the very least, by the lubricant formulation employed. Since, apparently, the lubricant can't be fully disclosed, a reference to "a current fully formulated commercial grease" should suffice. On the same note, any further clarification on the lubricant, base oil type, thickener concentration or additive package would go a long way in making the research replicable. The lubricant formulation is expected to shift these margins and likely also influences the effectiveness of the so called "protection runs". A different additive package might require less time to form tribo-layers, changing the impact of frequency effects, for example. Alternatively, a base oil with a different chemistry may have a similar viscosity but different surface wetting properties which might also affect the impact of frequency.

- x/2b value is used in the main parameter table but is seldom used elsewhere when referring to the tests. I would encourage authors to add the x/2b value alongside the angle as it is referenced in the text. Makes for a much easier read, rather than having to move back and forth back to the table. It would also be particularly useful in communicating the amplitudes of the protection runs.

- All wear mark figures are missing a reference scale length. Considering that there is no quantification of wear other than optical imagery, I think it would be quite relevant to add these. In figure 6, for instance, it is difficult to assess whether each of the two images are scaled identically.

- In Figure 7: The bearing torque appears quite strange even at 1000 cycles, which is the lowest cycle count in this figure. With a horizontal offset at the 0-torque horizontal line. I may be mistaken, but what would be expected is either parallelogram friction torque loops; such as the ones reported later in Figure 9 or a pre-rolling narrow diagonal slit. I find that the lack of an explanation on the shape in figure 7, combined with the lack of a healthy bearing torque loop at 1-10 cycles, negatively impact the clarity. Is this a product of backlash? It seems to be less apparent in figure 9, which suggests that this is the case. My suggestion here would be to address this in the text such that the reader is able to discern whether this is a product of the friction in the bearing, a result of the experimental setup or else.

- Building on the last point, discussion of "inertial forces" in line 229 suggests that the authors are not removing the acceleration torques from the curves reported as friction torques. This might explain the why some of the torque loops look odd, at least partially. It is also important to note for the implications of interpretation of figure 9. Clarification on whether the torques reported are indeed the direct sensor feed, or alternatively, have been processed to remove inertial effects would be a welcomed improvement in terms of clarity.

**Technical Corrections:**

- Line 124: *"A torque measurement is mounted to the pinion shaft"*. This sentence should be completed, for example: A torque measuring device is mounted to the pinion shaft.

- Line 247: *"Test ID V fits perfectly and supports the statement, that wear severity decreases with higher amplitudes. The test with the smallest amplitude diverges. It has a slightly lower characteristic."* I do not understand what is being said here: "slightly lower characteristic"? I am almost certain that the idea being communicated is that it has less pronounced wear but it should be rewritten for clarity.

- Line 244: *"It was also confirmed for high x/2b ratios by Schwack et al., that have seen similar (Schwack et al., 2020)."* Consider revising the writing of this sentence, "that have seen similar phenomena", or trends, for example.

---

## Author Comment (AC3)

**Supplement AC3 on RC1 to "Exploring Limiting Factors of Wear in Pitch Bearings of Wind Turbines with Real Scale Tests"**

Equation 1

$$\Delta M = (\ddot{\varphi}_1 - \ddot{\varphi}_2) \cdot 3 \cdot I_{OR} = 1.4 \; kN$$

Speed an Acceleration of Test ID III

[Figure]

Speed an Acceleration of Test ID IV

---

## Author Comment (AC4)

**Supplement AC4 on RC4 to
"Exploring Limiting Factors of Wear in Pitch Bearings of Wind
Turbines with Real Scale Tests"**

[Figure]

Figure 1: Friction torque over time of Test ID I and ID II

---

## Author Response (AR1)

**Author's response**

**Exploring Limiting Factors of Wear in Pitch Bearings of Wind Turbines with Real Scale Tests**

We would like to thank all reviewer for their in-depth reviews and your helpful questions and comments. Please find our answers and changes below.

The comments are in green and our answers in black. Changes to the text are in marked ins blue.

Karsten Behnke and Florian Schleich
January 2023

**Review 1**

In line 11 it says "The size of the bearings and the test parameters differ from other published test results for oscillating bearings." In this case, it should already be added in the abstract what the concrete differences are.

In our opinion it would exceed the abstract. The differences are manyfold and listed in the tables below. However, to be clearer the following part was added to the sentence.

*The size of the bearings and the test parameters differ from other published test results for oscillating bearings, where often scaled bearings are used.*

In my opinion, the statement that damage always occurs must be put into perspective (line 14). Only one lubricant was examined. Work by C. Schadow, S. Tetora and M. Grebe, for example, clearly shows the influence of the lubricant on the development of damage.

*Hence, no wear limits can be defined with the tested grease and within typical operating conditions of a wind turbine below which wear does not occur.*

x is defined as the complete travel path of the roller within the raceway (the term "half cycle" can be confusing) (line 43).

That's correct. We deleted the "half".

*A way to compare wear tests is the dimensionless x/2b ratio. It relates the travel of the rolling element during a  cycle (x) to the width of the Hertzian contact (2b).*

In line 71, a few more references should be added, e.g. from M. Grebe, C. Schadow or S. Tetora, who have already made these statements much earlier.

It is right, the other authors showed that tests with "small" bearings give a good understanding of principial wear mechanism. However, the key statement of this sentences for me refers to the scaling effect. To my knowledge Stammler was the first who published wear test results with an original size blade bearing and compared the results to smaller ones.

In line 95, the publication by Grebe from 2006 or 2008 should be listed, who already describes these effects in this way.

You are right, they are added.

In line 103, the work of C. Schadow and S. Tetora from Magdeburg should be mentioned.

You are right, they are added.

In Table 2, the roughness of the treads would be an interesting parameter.

It is added in the table.

*Roughness (Ra)0.8        μm*

In line 129, the size of the contact surface should be added as another important parameter that can be calculated using Hertzian formulae.

It is added in the table.

| Test ID | Contact size in mm |
|---------|--------------------|
| I | 38.5 |
| II | 25.1 |
| III | 38.5 |
| IV | 38.5 |
| V | 29.8 |
| VI | 38.5 |

Line 189 and 214: The fact that damage can occur after only a few cycles was first published in *Grebe, Feinle, Hunsicker: Various Influence Factors on the Development of Standstill Marks (False-Brinelling Effect), ELGI-Meeting Lisbon, 2008.*
Likewise, the influence of the frequency (line 231).
The reference is added in the paper.

In line 234, 246 and 305, the corresponding x/2b ratio should be given in addition to the angle.
It is added in the paper.

**Review 2**

The relation between wear and torque is not established sufficiently.

To explain the relationship better, the following part was added to the paper. It was added to line 212.

*For an undamaged bearing Dahl describes the characteristic torque hysteresis curve for an oscillation (Dahl, 1968). In Figure 7 it is possible to see similar curves, but thereby the friction torque increases with more cycles, and it rises faster for the higher load. The peak is reached in the turning points. The raising torque is caused by the surface changes due to wear. Higher roughness values result in a higher friction. Furthermore, the wear mechanism like abrasion dissipate energy, which in turn lead to a higher friction torque (Fouvry et al., 2003). The trend of wear development of these tests as shown in Figure 6 and Figure 7 confirm Bartschat's and Wandel's conclusion that under certain circumstances wear develops quite fast (Wandel S., and Bartschat B., 2021).*

Some related previous work such as Yang et al.1 considered wear volume as a wear index. It would be beneficial to express the priority of the wear index in this study compared to previous work such as Yang et. al.

Yang et al. test bearings with focus is on rolling contact fatigue. The volume or weight of material that burst out of the ring is significant more compared to the material loss in our tests. In our tests we have indents with a few µm depth. For such small changes we do not have a sufficient method to determine the volume or the weight of material. Please do not forget that the rings weight about 1,000 kg.

The paper doesn't clearly specify the wear marks happen in which rings (inner or outer) and the reason and discussion on it.

The difference between both rings in contact pressure is less than 3% (e.g., inner Ring 2.55 GPa and outer ring 2.48 GPa)

*The contact pressure refers to the most highly loaded row at the inner ring. In the following only pictures of the inner ring are shown.*

Mesh independency of the finite element results are not stated.

The results are less dependent on the element size and mesh density as it would be the case for a detailed modelled bearing with internal contacts. Hence, the used modelling approach is less affected by that. Furthermore, typically the kind of extracted data that is affected by the element size and the mesh density are resulting stresses in the structural components. The presented results do not include any postprocessed stresses.

In the finite element modeling, tolerances and internal dimensions between the bearing rows are not considered. It is recommended to put some notes about the reason of not considering in the respective section.

The following is added to the text.

*Tolerances and manufacturing errors are not considered in the bearing model as no information on these parameters are available for each individual bearing and they are assumed to be negligible for the analysis of the global ring deformation behavior and the global load distribution.*

Most frequent resulting moment according to Figure 5 is less than 2.0 MNm. Clarified the sentence "**A frequently resulting moment is 2.5 MNm**" in line 174.

The next sentences refer explicitly to 2.5MNm. Hence, it is introduced as one resulting moment, which occurs oftentimes or "frequently". It is right that the "most frequent" moments are less than 2.0MNm, but it does not match to the context.

In line 88 there are repeated "**to**", and one should be cleared.
It is deleted.

**Review 3**

One of the main findings stated in the abstract is the non-existence of "wear limits". This statement should be conditioned, at the very least, by the lubricant formulation employed. Since, apparently, the lubricant can't be fully disclosed, a reference to "a current fully formulated commercial grease" should suffice. On the same note, any further clarification on the lubricant, base oil type, thickener concentration or additive package would go a long way in making the research replicable. The lubricant formulation is expected to shift these margins and likely also influences the effectiveness of the so called "protection runs". A different additive package might require less time to form tribolayers, changing the impact of frequency effects, for example. Alternatively, a base oil with a different chemistry may have a similar viscosity but different surface wetting properties which might also affect the impact of frequency.

Your right. The following sentence from the abstract was edited, to make this point clear.

*Hence, no wear limits can be defined with the tested grease and within typical operating conditions of a wind turbine below which wear does not occur.*

x/2b value is used in the main parameter table but is seldom used elsewhere when referring to the tests. I would encourage authors to add the x/2b value alongside the angle as it is referenced in the text. Makes for a much easier read, rather than having to move back and forth back to the table. It would also be particularly useful in communicating the amplitudes of the protection runs.

We tried to avoid the x/2b-values as much as possible in the text. The focus of this work was not on the x/2b-ratio, since only one bearing size was tested and there was no need for scaling. x/2b is useful for comparing different bearing sizes. Two (or theoretically an endless number of) completely different combinations of pressure and angle could lead to the same x/2b, which means that the same x/2b ratio could lead to two different wear marks. So just the x/2b ratio does not give a distinct overview of the test setup.

However, we added it to the text for easier read. This is already done according to RC1.

All wear mark figures are missing a reference scale length. Considering that there is no quantification of wear other than optical imagery, I think it would be quite relevant to add these. In figure 6, for instance, it is difficult to assess whether each of the two images are scaled identically.

That true unfortunately we missed that point. All wear marks were investigated with a laser scanning microscope as well, to analyze the roughness, size, and the depth of the wear marks. A calculation of the removed volume does not deliver satisfying results. The size of the wear marks as well as the curvature of the raceways impede to take reliable and repeatable pictures.

Pictures with a scale can be provided later.

In Figure 7: The bearing torque appears quite strange even at 1000 cycles, which is the lowest cycle count in this figure. With a horizontal offset at the 0-torque horizontal line. I may be mistaken, but what would be expected is either parallelogram friction torque loops, such as the ones reported later in Figure 9 or a pre-rolling narrow diagonal slit. I find that the lack of an explanation on the shape in figure 7, combined with the lack of a healthy bearing torque loop at 1-10 cycles, negatively impact the clarity. Is this a product of backlash? It seems to be less apparent in figure 9, which suggests that this is the case. My suggestion here would be to address this in the text such that the reader is able to discern whether this is a product of the friction in the bearing, a result of the experimental setup or else.

You are right the curves in figure 7 looks unusually compared to a theoretical friction curve. Since the initial torque curve (after a few cycles) looks like the ones shown in figure 7 (with lower peak values), we would assume that it is an issue with the backlash of the gearbox. The gearbox itself has three planetary stages. Furthermore, the backlash of the pinion towards the bearing adds up, too. To clarify this point in the text we added the following to the text:

*For an undamaged bearing e.g. Dahl describes the characteristic torque hysteresis curve for an oscillation (Dahl, 1968). The backlash between the driving pinion and the bearing ring will influence the torque and they might look different than a theoretically curve with an ideal drive. However, in Figure 7 it is possible to see similar curves, but thereby the friction torque increases with more cycles, and it rises faster for the higher load.*

Building on the last point, discussion of "inertial forces" in line 229 suggests that the authors are not removing the acceleration torques from the curves reported as friction torques. This might explain the why some of the torque loops look odd, at least partially. It is also important to note for the implications of interpretation of figure 9. Clarification on whether the torques reported are indeed the direct sensor feed, or alternatively, have been processed to remove inertial effects would be a welcomed improvement in terms of clarity.
The highest accelerations of the tests IDIII and IDIV (from figure 9) are 0.17 rad/s² and 0.004 rad/s².

Speed an Acceleration of Test ID III

[Figure]

Speed an Acceleration of Test ID IV

[Figure]

Of all tests this is the highest difference in speed and acceleration. Hence, it is the worst case. The inertia of one outer ring is 2.76e4kg*m². If the bolts and balls are neglected and just 3 outer rings are considered, the following equation gives the difference caused by inertia between the friction torques.

$$\Delta M = (\ddot{\varphi}_1 - \ddot{\varphi}_2) \cdot 3 \cdot I_{OR} = 1.4 \; kN$$

Assuming a nominal torque of 25kN (compare figure 9) it is less than 6% difference. The speed and the acceleration have a sinusoidal profile. Hence, most of the time the difference of both curves is less than 6%. The controller, converter, motor, and the gearbox influence the theoretical acceleration and hence the inertia force slightly. Therefore, and for the small overall influence, we decided to keep the original measurement data.

Line 124: *"A torque measurement is mounted to the pinion shaft"*. This sentence should be completed, for example: A torque measuring device is mounted to the pinion shaft.
It is changed to:

*The pinion shaft is equipped with a torque measurement device.*

Line 247: *"Test ID V fits perfectly and supports the statement, that wear severity decreases with higher amplitudes. The test with the smallest amplitude diverges. It has a slightly lower characteristic."* I do not understand what is being said here: "slightly lower characteristic"? I am almost certain that the idea being communicated is that it has less pronounced wear but it should be rewritten for clarity.
It is changed to:

*It  has less pronounced wear.*

Line 244: "*It was also confirmed for high x/2b ratios by Schwack et al., that have seen similar (Schwack et al., 2020).*" Consider revising the writing of this sentence, "that have seen similar phenomena", or trends, for example.
It is changed to:

*It was also confirmed for high x/2b ratios by Schwack et al., that have seen similar* trends *(Schwack et al., 2020).*

**Review 4**

Figure 3. y axis should be between 30000 and 70000 or smaller to see the differences among loads. Also, will the gravity of the upper bearing and surrounding structures play a role on the load differences observed?

In case it would be of interest how the load distributes over the circumference and what might be the differences of minimum and maximum ball load on each raceway, a scale of the y axis between 30000 and 70000 might be more sufficient. However, as the focus of figure 3 is to point out how the loads distribute on both rows of the bearings, it is not beneficial to change the scale of the y-axis. The differences among the loads (between both rows!) can be seen clearly also in this kind of the plot.

Several simulations have been carried out during the development of the FE model. Comparisons of displacements and ball forces between a simulation with gravity and a simulation without gravity reveals that the contribution of the masses to the resulting deformations and loads is almost zero. In relation to the applied forces by the hydraulic actuators the gravitational loads are very small. Nevertheless, in all performed simulations the gravitational forces are considered, but due to the negligible effect this detail is not mentioned in the paper.

Is there a different way to present the results in a qualitive way? For an example, Figure 7 (effects of contact pressure on friction torque over time), is it possible to generate a 2D plot with x-axis being number of cycles and y-axis being torque or its percentage? Both cases with 2GPa and 2.5GPa contact pressure can be presented on the same plot. There are many figures that can be converted this way. Would authors consider this?

Yes, of course there is a different way, but it is a trade of. One piece of information is always missing, e.g. if showing the torque over the entire time the torque for one cycle isn't visible anymore, and hence the information at which angle the maximum torque occurs.
Please see the following figure 1, which shows the torque of Test IDI and Test IDII (both for 5,000 cycles).

[Figure]

Figure 1: Friction torque over time of Test ID I and ID II

Vast amount of information of the measurement are presented. But not all test results are well discussed. For example, Figure 12, given the same x/2b ratio, the wear formation was not impacted by the bearing size. This is an important finding. Authors might consider adding discussion on x/2b here.

Figure 12 gives evidence for the scaling with x/2b. But for a complete discussion on the x/2b ratio it is probably to less. Therefore, further information would be necessary like cleaning process, grease, test

speed, cycles, and others.
However, this paper refers back to previous research conducted on smaller bearings, so at least a clue if the results are comparable is useful.

Suggestions on protection run is very interesting and its effect on pitch bearing wear is promising. As authors mentioned, its implementation will require further research, in particular for its impact or interactions with energy output and supervisory controls. Authors discussed some of these aspects in the conclusion but, in my opinion, it should be expanded and incorporated within section 5.2.
Within this project the protection runs just show how to avoid wear and if it is possible. Further aspects are considered in other projects in more detail. We would like to refer to the HAPT project and the iBAC project:

https://www.enargus.de/detail/?id=698541
https://www.enargus.de/detail/?id=1143011

In both projects the mentioned aspects are discussed in further detail.

Lastly, authors should consider merging paragraphs with only a few sentences.
Done in the reviewed paper.